# Which Are the Best Site and Stand Conditions for Silver Fir (*Abies alba* Mill.) Located in the Carpathian Mountains?

Lucian Dinca [1], Mirabela Marin [1], Vlad Radu [2], Gabriel Murariu [3], Romana Drasovean [3], Romica Cretu [3], Lucian Georgescu [3] and Voichița Timiș-Gânsac [4,*]

1 "Marin Drăcea" National Institute for Research and Development in Forestry, 13 Closca Street, 500040 Brasov, Romania; dinka.lucian@gmail.com (L.D.); mirabelamarin@yahoo.com (M.M.)
2 "Marin Drăcea" National Institute for Research and Development in Forestry, 73BIS Calea Bucovinei Street, 725100 Campulung Moldovenesc, Romania; vlad.radu2@gmail.com
3 Chemistry, Physics, and Environment Department, Faculty of Sciences and Environment, "Dunărea de Jos" University of Galaţi, No. 111 Street Domnească, 800201 Galaţi, Romania; gmurariu@ugal.ro (G.M.); romana.drasovean@ugal.ro (R.D.); romica.cretu@ugal.ro (R.C.); lucian.georgescu@ugal.ro (L.G.)
4 Department of Forestry and Forest Engineering, University of Oradea, 410048 Oradea, Romania
* Correspondence: timisvoichita@yahoo.com

**Abstract:** Silver fir (*Abies alba* Mill.) is one of the most valuable and productive tree species across European mountains, that accomplish multiple economic, protective and ecologic functions. Alongside spruce (*Picea abies* (L.) Karst) and beech (*Fagus sylvatica* L.), silver fir is a characteristic species for the Romanian Carpathians. Although silver fir tree is recommended for the diversification of forests in order to increase the resistance to climate change, it is very sensitive to climatic excesses, especially those that proceed rapidly. Therefore, the aim of this study is to investigate both the environmental conditions and stand characteristics of fir from five mountain ranges of the Romanian Carpathians. The study is based on data recorded over a period of 10 years (1990–2000). As such, a total of 77,251 stands that occupy 211,954 hectares have been investigated in regard to silver fir behaviour. MATLAB scripts were used for analysing consistent data volumes as well as the impact of eight factors on the silver fir productivity (altitude, field aspect, field slope, soil type, participation percentage, road distance, structure and consistency). Our analysis has revealed that higher silver fir productivity is found at altitudes of up to 1200 m, on mid and upper slopes, on NW field aspects, on eutric cambisols and dystric cambisols, with a 10–20% participation in stand composition and in relatively-even aged stands with a full consistency. This study offers valuable insights for forest managers that require comprehensive information in adopting effective strategies to enhance forest resilience under climate change.

**Keywords:** silver fir; Romanian Carpathians; stand productivity; forest sustainability

## 1. Introduction

Considering the essential role of forests in providing many ecosystem services, the scientific community has been interested in the last years in evaluating the connections between forest diversity and their productivity [1,2]. Climatic changes influence forest productivity both directly (by altering temperature and rainfall patterns) [3,4] as well as indirectly (through the action of different harmful and pathogen agents) [5–7]. Even though forest productivity at European level has followed a growing trend, forecasts regarding the growth of extreme phenomena in intensity and frequency will alter water resources and, implicitly, forest productivity through water-limiting [2,5,8,9].

Silver fir (*Abies alba* Mill.) is considered one of the most valuable and productive species from mountainous areas of Europe, due to its ecologic, economic and soil protection

functions [10,11]. Due to its high productivity, silver fir is recommended in forest diversification in order to enhance the resistance towards climatic changes [12] as well as for protecting carbon stocks from European forests during the following centuries [13,14]. European forests, comprised mainly of silver fir, common beech (*Fagus sylvatica* L.) and Norway spruce (*Picea abies* (L.) Karst) and widespread on over 10 million hectares, are very sensitive towards climatic changes, especially those that occur more rapidly and do not allow trees to adapt over time [15,16]. Silver fir is one of the species with high requirements towards stational conditions, having high requests towards soil and air humidity [17–19]. Very sensitive towards climatic excesses, especially during youth, seedlings suffer from drought, insolation, late frosts, excessive frost, cold and dry winds and soil and air dryness. Hence, during the last decades, silver fir faced a European decline due to atmospheric pollution and climatic changes, that led to a decrease up to 80% within the Carpathians [20,21].

Romania owns an important part of Europe's natural resources. Some of the largest compact forests from the temperate area are located in our country, together with the oldest forests from Europe [22,23]. Due to the fact that mountain areas occupy 23% of the total global forest surface, a sustainable management of these ecosystems is emphasized in the 2030 Agenda for Sustainable Development [24,25]. Romania holds over 50% of the Carpathians' surface, considered the largest mountain chain from Central Europe [26]. Silver fir occupies 5% of their afforested surface, being widespread in the Eastern Carpathians, Southern Carpathians, Curvature Carpathians, Apuseni and Banat Mountains [19].

If we consider silver fir's ecologic and economic importance, as well as its sensibility towards climate excesses, numerous international and national studies focused on assessing the influence of stational conditions on the specie's development. As such, some authors [27] pursued to identify the role of soil properties on the growth of silver fir and have shown that soil parameters influence the growth of this species. Furthermore, Pinto [28] mentions that, besides soil properties, climate and stand composition also control the growth and development of silver fir. On the other hand, Bosela's study [29] has shown that the specie's genetic diversity has an important role in the silver fir's growth and resistance towards climatic changes. Mohytych [30] intended to identify the most favourable conditions for the regeneration of this species. In addition, research by Lasch and Sperlich [9,31] have emphasized that silver fir's productivity will decrease in the following years as a consequence of climatic changes.

Studies conducted in Romania have shown that some resinous species, among which *Abies alba* Mill., are affected by *Viscum album* ssp. abietis. Mistletoe attack is among the factors that degrades old silver tree stands [7,17,32]. The study performed by Dincă and his collaborators [33] shows that silver fir stands located on the north slopes of Southern Carpathians evince superior productivity compared to the south slopes. The study realized by Tudoran [34] emphasizes that silver tree record height growths up to the age of 80, reaching a maximum between 50 and 60 years after which, trees accumulate growths in diameter. Dendrochronological growths performed by Kern and Popa [35] have shown that the radial growth of silver tree is positively correlated with both abundant precipitation from spring and summer and mild winters, as well as with temperatures recorded during summer months and early winter [36]. On the other hand, studies conducted by Dinulică shown that a radial growth is influenced by wind. However, an important role in mitigating the impact of climate changes is posed by the genetic variation of local, regional and national silver fir populations, as mentioned by Teodosiu [19,37].

Therefore, the purpose of this study is to identify the key characteristics (stational or stand) that lead to the occurrence of superior productivity classes in silver fir from the five Romanian Carpathians Mountain chains. In order to reach this goal, we have established two specific objectives, namely: (1) evaluating stational conditions, namely altitude, field aspect, slope and soil type in each of the five areas; and (2) evaluating stand conditions, namely participation percentage, distance from the road, stand structure and consistency within the five mountain chains. The research's outcomes are useful for forest managers in

establishing the most adequate management measures that can increase the resilience of silver fir in the context of different changes (climatic and environmental).

Climate factors are important for the specific composition of forests, while stational conditions can determine sensible changes at smaller scales. Amongst these, the topographic ones influence both forest structure, species distribution and diversity as well as floristic composition [38].

The studies performed by Sidor and his collaborators in Banat (Romania), have shown that the average surface growth for silver tree presents almost double values, compared with Scots pine and larch. Furthermore, silver fir had a positive response to the precipitations from the beginning of the vegetation season [39]. The positive influence of temperature from the cold season on the radial growth of silver fir was emphasized in other areas from the Romanian Carpathians as well as from Europe [36,40,41].

Studies focused on silver fir behaviour in south-east Carpathians revealed both a negative correlation between radial growth and temperature, as well as a positive correlation between radial growth and precipitation during the growth season.

The aim of this study is to investigate both the environmental conditions and stand characteristics of fir from five mountain ranges of the Romanian Carpathians.

The working hypotheses were:

(a) there is a strong connection and a certain dependence between the production classes from the forest areas of different mountain massifs?

(b) are the productivity classes are influenced by a series of parameters?

As the preliminary investigation the distributions of the studied parameters was a priori performed, it was observed that the obtained results do not always respect the conditions of normal distribution.

## 2. Materials and Methods

### 2.1. Study Area

The study area comprehended all silver fir stands from the five Romanian Carpathians Mountain chains, namely: Apuseni Mountains (AM), Banatului Mountains (BM), Eastern Carpathians (EC), Curvature Carpathians (CC) and Southern Carpathians (SC), that comprises silver fir in their composition (Figure 1).

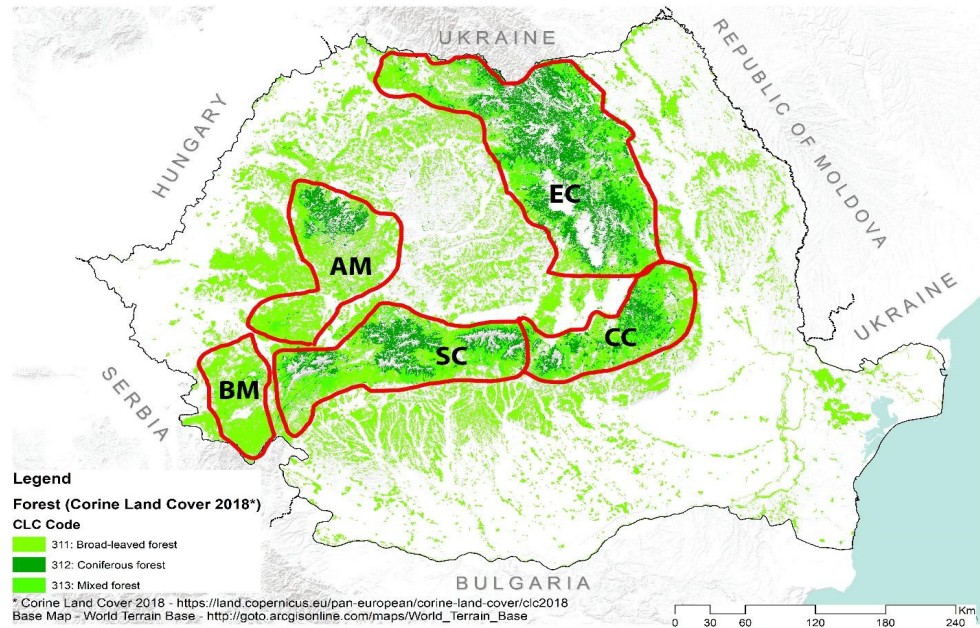

**Figure 1.** Carpathian Mouintains: Apuseni Mountains (AM), Banatului Mountains (BM), Eastern Carpathians (EC), Curvature Carpathians (CC), Souuthern Carpathians (SC).

The analysed silver fir stands were: in the AM: 4203 stands and 10,143 ha; in the BM: 4185 stands and 11,980 ha; in the EC: 37,385 stands, 103,020 ha; in the CC: 15,304 stands, 41,778 ha; in the SC: 16,174 stands, 45,024 ha. Total Carpathians: 77,251 stands, 211,945 ha.

### 2.2. Data Collection and Methods

For each stand, we used information from forest management plans for 1990–2000 period. Data regarding productivity classes were extracted in order to analyse the influence of environment factors (altitude, field aspect, field slope and soil type) and stand factors (participation percentage, road distance, structure and consistency) on the productivity of silver fir stands. Data from forest management plans for 1990–2000 period were used [42]. This period was chosen as it precedes the massive transmission of forests towards previous owners, in which case, complete forest management plans are no longer requested.

The altitude was measured on the field with GPS devices, the field aspect was established based on the area's mapping. The fields slope was measured with a hypsometer. The soil type was determined by realizing soil profiles, gathering samples and analysing them in specialty laboratories. The participation percentage in the stand's composition, as well as stand structure and consistency were established visually. Road distance and altitude was analysed from maps with level curves.

In the case of even-aged stands, the site index can be established based on the average height and age. For uneven-aged stands, this value can be determined by the average height that corresponds to the average diameter of 50 cm. The site index reports the height of dominant and co-dominant trees in a stand at a base age. Thus, it is used to measure the site productivity, to determine site management options and to describe the potential of forest trees to grow at a particular site. In our case, the site index was calculated for each stand.

In order to determine the optimal stational and stand conditions for silver fir, we have grouped all values for the first and second productivity classes (CP) and afterwards compared them with the sum of the other values (belonging to classes productivity 3 + 4 + 5).

### 2.3. Statistical Calculations

The data used in this study was processed with StatSoft Statistica and MATLAB statistical analysis scripts which is specific for a consistent volume of data [43]. In order to verify the distribution type in the case of all parameters we applied the Kolmogorov-Smirnov tests. As it is already known, the Kolmogorov-Smirnov test is used for testing large sets of data [44].

The parametric ANOVA and Non-Parametric Tests, such as Kruskal-Wallace Test, were applied simultaneously to highlight the simultaneously to highlight the consistency of the addiction investigation.

Moreover, the comparison T statistical tests of Fisher two tails type were applied for observing significant statistical differences between the values of the studied parameters. In the case where the p threshold parameter's value was under 0.05, we have considered the differences between averages as significant and added an observation in the presentation's text. If the threshold's value is superior to the imposed value, the observed differences were considered as insignificant.

## 3. Results

### 3.1. Altitude

Regarding the distribution of altitude for the silver fir stands for each of the five area (Table 1; Figure 2) we can observe that, in AM, EC, SC and Total, the high productivity silver tree is founded at altitudes between 857–1042. In contrast, in the BM, the upper silver fir productivity class (CP) occurs at high altitudes (over 900 m), while in the case of CC, high attitudes are responsible for the lower-class fir (996 m). For the Total of the five mountain ranges analysed, the upper-class fir is found at low altitudes (800–900 m).

**Table 1.** Average altitude (m) for silver fir trees located in the Carpathians.

| Mountain Area | Altitude (m) for Productivity Classes | | | | |
|---|---|---|---|---|---|
| | **1** | **2** | **3** | **4** | **5** |
| Apuseni Mountains = AM | 870 | 1020 | 1054 | 1035 | 1055 |
| Banatului Mountains = BM | 939 | 915 | 845 | 864 | 879 |
| Eastern Carpathians = EC | 886 | 940 | 962 | 950 | 1010 |
| Curvature Carpathians = CC | 1011 | 1003 | 996 | 1034 | 1026 |
| Southern Carpathians = SC | 1042 | 1063 | 1096 | 1105 | 1110 |
| Total Carpathians | 857 | 917 | 972 | 986 | 1033 |

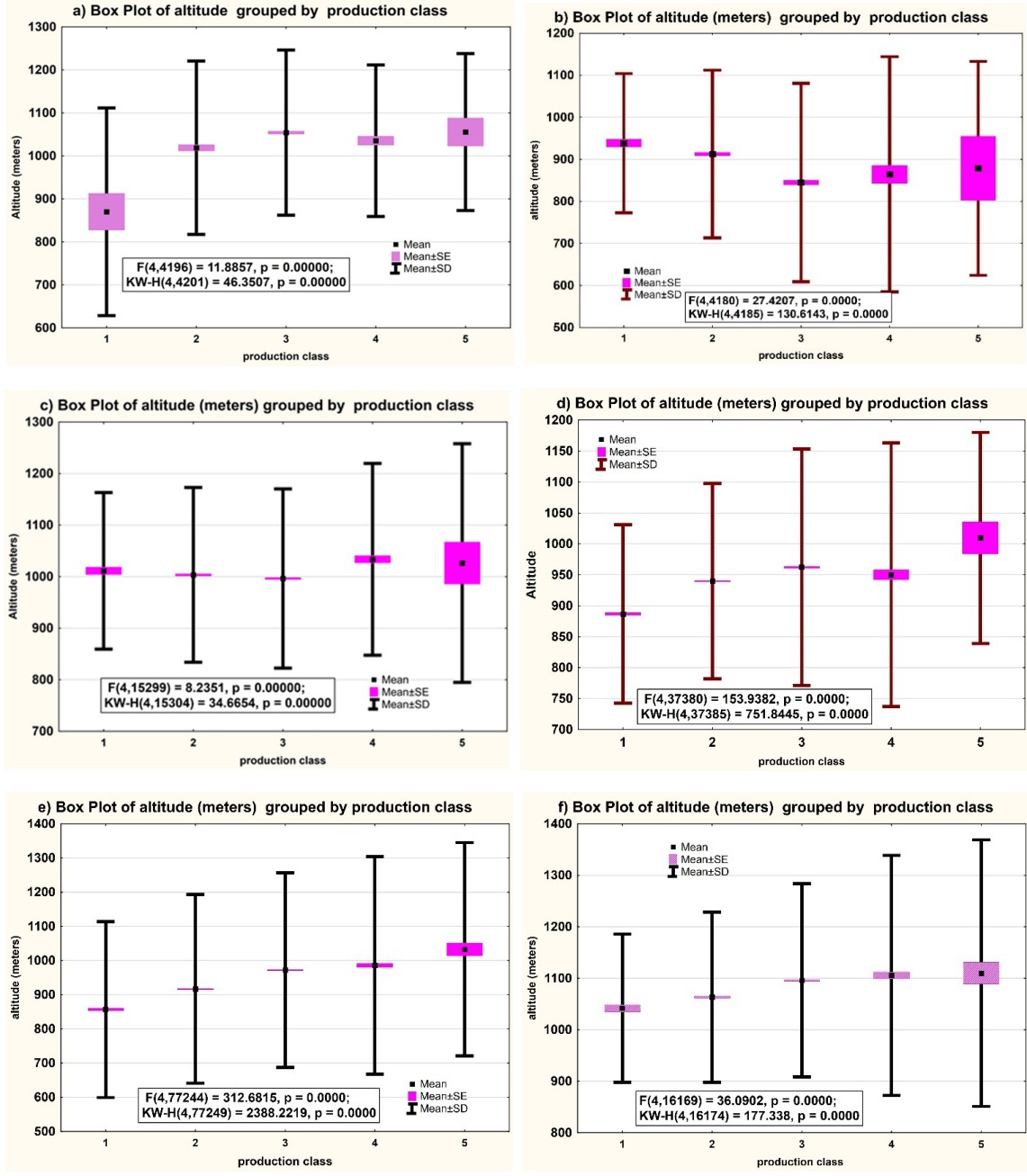

**Figure 2.** (**a**) Altitude AM. (**b**) Altitude BM. (**c**) Altitude EC. (**d**) Altitude CC. (**e**) Altitude SC. (**f**) Altitude TOTAL.

In this sense, we have applied comparison T statistical tests of Fisher type which shown that the height distribution for the superior class has a significantly lower average than the other distributions ($p = 0.0001$). These tests have proved significant differences for both T one tail tests as well as for two tails.

We investigated the existence of significant differences in the average values obtained for the series of parameters related to each production class. We were able to demonstrate by using Fisher-type tests that there were no significant differences.

In principle—according to the reference [26], the preliminary evaluation of the distribution of the studied parameters was performed. The obtained distributions do not always respect the conditions of normal distribution. Under these conditions, parametric and non-parametric ANOVA analyses were applied.

### 3.2. Field Aspect

Silver fir has the largest propagation in the Romanian Carpathians on the NW field aspect (Figure 3a). This phenomenon is more significant in superior and average productivity stands. The Curvature Carpathians do not record field aspect differences, probably due to their shape (the curvature comes from their shape, with a transition from N-S to E-W). The same context is present in Carpathians Total (Figure 3b).

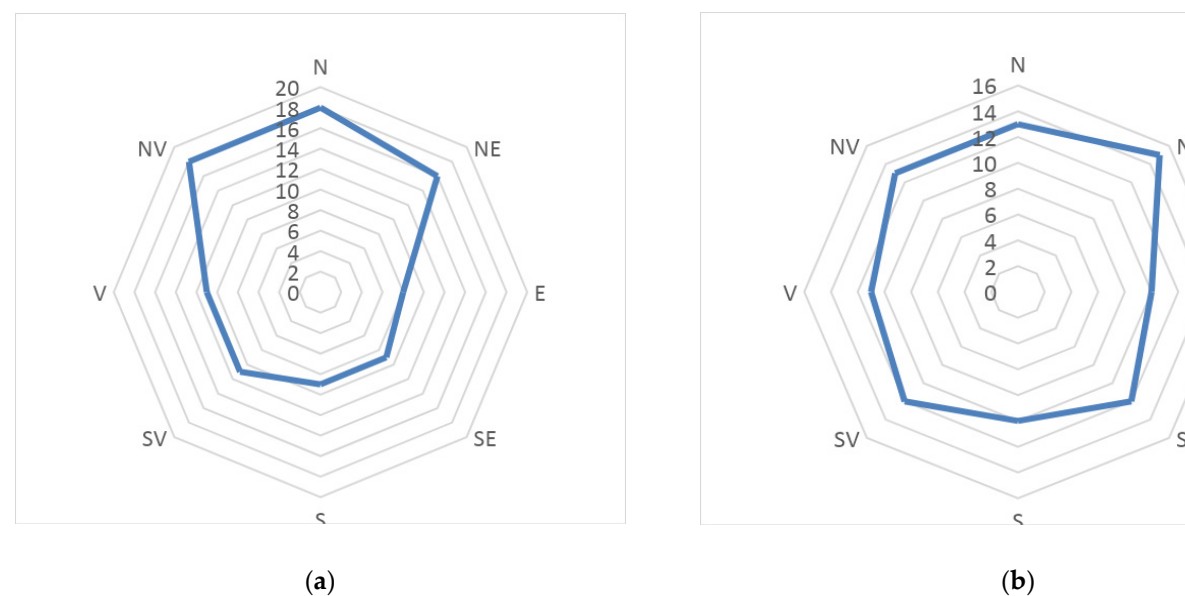

(**a**)　　　　　　　　　　　　　　　　　　　　　　(**b**)

**Figure 3.** (**a**) AM-field aspect for classes 2 prod. (**b**) Total-Carpathians field aspect.

Analysing the data from Table 2 see can see a preference for shadowed field aspects (for AM, EC and SC). In the case of the BM, on the other hand, the shaded exposures host fir trees of lower productivity classes, while for the CC, there are no differences in exposures, probably due to their shape (the curvature comes from their shape, with a transition from N-S direction to E-V).

### 3.3. Field Slope

The field slope on productivity classes is given in Figure 4, for each of the five areas. Analysing the figures below, we distinguish the fact that significant differences between the productivity class and the slope of the land appear at the lower productivity classes (CP4 and CP5) in each of the five mountain ranges analysed. The same situation is encountered in the case of Total Carpathians where productivity classes 4 and 5 show the most significant differences.

**Table 2.** The distribution of silver fir trees from the Carpathians on altitudes (%).

| Mountains | Prod Class | N | NE | NV | E | SE | S | SV | V |
|---|---|---|---|---|---|---|---|---|---|
| AM | 1 | 13 | 13 | 22 | 12 | 16 | 9 | 9 | 6 |
| | 2 | 18 | 16 | 18 | 8 | 9 | 9 | 11 | 11 |
| | 3 | 16 | 15 | 17 | 9 | 11 | 7 | 13 | 12 |
| | 4 | 17 | 14 | 15 | 9 | 11 | 10 | 11 | 13 |
| | 5 | 7 | 3 | 23 | 6 | 13 | 23 | 19 | 6 |
| | Total | 17 | 15 | 17 | 9 | 10 | 7 | 13 | 12 |
| BM | 1 | 9 | 17 | 18 | 9 | 16 | 10 | 12 | 9 |
| | 2 | 13 | 12 | 15 | 10 | 14 | 13 | 15 | 8 |
| | 3 | 14 | 13 | 15 | 9 | 13 | 12 | 13 | 11 |
| | 4 | 16 | 12 | 14 | 8 | 13 | 16 | 9 | 12 |
| | 5 | 27 | 9 | - | 9 | - | 28 | 18 | 9 |
| | Total | 14 | 13 | 15 | 9 | 14 | 12 | 13 | 10 |
| EC | 1 | 14 | 16 | 16 | 11 | 13 | 8 | 12 | 10 |
| | 2 | 14 | 14 | 17 | 9 | 13 | 10 | 12 | 11 |
| | 3 | 14 | 14 | 16 | 10 | 13 | 10 | 12 | 11 |
| | 4 | 12 | 12 | 15 | 14 | 11 | 15 | 11 | 10 |
| | 5 | 18 | 9 | 12 | 14 | 12 | 12 | 18 | 5 |
| | Total | 14 | 14 | 17 | 9 | 13 | 10 | 12 | 11 |
| CC | 1 | 10 | 14 | 17 | 9 | 16 | 9 | 13 | 12 |
| | 2 | 12 | 15 | 17 | 9 | 12 | 10 | 14 | 11 |
| | 3 | 15 | 15 | 14 | 9 | 11 | 12 | 12 | 12 |
| | 4 | 14 | 12 | 15 | 9 | 11 | 12 | 11 | 16 |
| | 5 | 6 | 7 | 13 | 7 | 19 | 19 | 13 | 16 |
| | Total | 14 | 15 | 15 | 9 | 11 | 11 | 13 | 12 |
| SC | 1 | 10 | 17 | 16 | 12 | 12 | 10 | 11 | 12 |
| | 2 | 13 | 16 | 19 | 12 | 10 | 8 | 11 | 11 |
| | 3 | 12 | 17 | 19 | 10 | 10 | 7 | 12 | 13 |
| | 4 | 11 | 13 | 22 | 12 | 10 | 7 | 12 | 13 |
| | 5 | 14 | 12 | 26 | 8 | 6 | 9 | 10 | 15 |
| | Total | 13 | 16 | 19 | 11 | 10 | 7 | 12 | 12 |
| TOTAL | 1 | 13 | 16 | 16 | 11 | 13 | 8 | 12 | 11 |
| | 2 | 13 | 15 | 17 | 10 | 12 | 10 | 12 | 11 |
| | 3 | 14 | 15 | 16 | 9 | 11 | 10 | 13 | 12 |
| | 4 | 13 | 12 | 18 | 11 | 11 | 11 | 12 | 12 |
| | 5 | 13 | 10 | 21 | 9 | 9 | 13 | 13 | 12 |
| | Total | 13 | 15 | 17 | 10 | 12 | 10 | 12 | 11 |

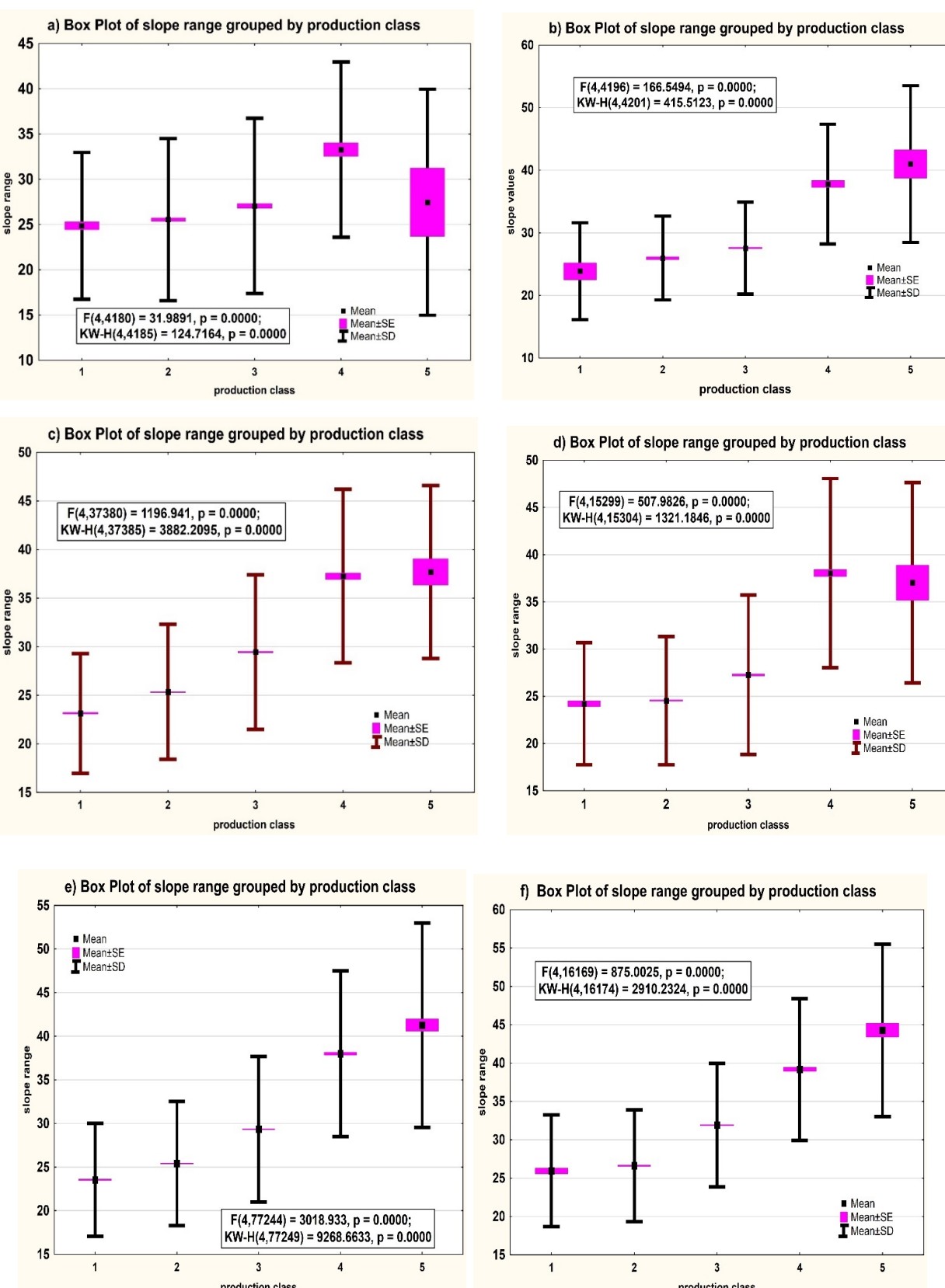

**Figure 4.** (**a**) Field slope AM. (**b**) Field slope B. (**c**) Field slope EC. (**d**) Field slope CC. (**e**) Field slope SC. (**f**) Field slope Total.

Low productivity silver firs are located in all cases on fields with a very high slope (27–44%) (Table 3). In this regard, we have applied T statistic tests of Fisher type and proved that height distribution on superior classes has a significantly lower average than the other distributions ($p = 0.0001$). These tests showed significant differences for both one tail and two tails *T* tests.

**Table 3.** Average slope for silver firs located in the Carpathians.

| Mountain Area | Slope (%) for Productivity Classes | | | | |
|---|---|---|---|---|---|
| | **1** | **2** | **3** | **4** | **5** |
| Apuseni Mountains = AM | 23.88 | 25.97 | 27.56 | 37.80 | 41.00 |
| Banatului Mountains = BM | 24.85 | 25.55 | 27.05 | 33.27 | 27.45 |
| Eastern Carpathians = EC | 23.14 | 25.34 | 29.46 | 37.25 | 37.70 |
| Curvature Carpathians = CC | 24.21 | 24.54 | 27.26 | 38.04 | 37.03 |
| Southern Carpathians = SC | 25.94 | 26.63 | 31.91 | 39.17 | 44.27 |
| Total Carpathians | 23.54 | 25.41 | 29.33 | 38.01 | 41.27 |

*3.4. Soil Type and Subtype*

The graphical representation of the main soil types and subtypes is shown in Figure 5. As we can see, there are no significant differences between the 2nd and 3rd productivity classes in any of the studied mountain areas. We can observe that calcic eutric cambisol is largest in BM (9%), while is very few SC (under 1%). Entic podzol is low at BM and more at EC (9%). For Total Carpathians, eutric cambisol and dystric cambisol occupy the same area (38%), with biggest differences between the two soils at AM, where dystric cambisol is larger than eutric cambisol (41% comapred with 28%).

The distribution of silver fir by soil types and subtypes for all five areas is shown in Table 4. Due to the large number of soil subtypes, we have taken into account only the 2nd (CP2) and 3rd productivity classes (CP3) as well as 3 soil types (eutric cambisol, dystric cambisol, entic podzol) + 2 subtypes (calcic eutric cambisol and lythic dystric cambisol), (according to the WRB soil classification).

**Table 4.** Soil type for silver fir stands located in the Carpathians.

| Mountain Area | Soil (%) | | | | | |
|---|---|---|---|---|---|---|
| | **Eutric Cambisol** | **Calcic Eutric Cambisol** | **Dystric Cambisol** | **Lythic Dystric Cambisol** | **Entic Podzol** | **Other Soils** |
| Apuseni Mountains = AM CP2 | 35 | 3 | 43 | 1 | 0 | 18 |
| AM-CP3 | 26 | 5 | 43 | 4 | 0 | 22 |
| AP-Total | 28 | 5 | 41 | 5 | 3 | 18 |
| Banatului Mountains = BM CP2 | 37 | 11 | 41 | 0 | 0 | 11 |
| BM-CP3 | 41 | 8 | 34 | 1 | 0 | 16 |
| BM-Total | 38 | 9 | 39 | 1 | 0 | 13 |
| Eastern Carpathians = EC CP2 | 43 | 1 | 44 | 3 | 0 | 9 |
| EC-CP3 | 30 | 0 | 37 | 12 | 2 | 19 |
| EC-Total | 38 | 1 | 42 | 5 | 9 | 5 |
| Curvature Carpathians = CC CP2 | 48 | 0 | 34 | 2 | 1 | 15 |
| CC-CP3 | 39 | 0 | 44 | 7 | 2 | 8 |
| CC-Total | 42 | 0 | 38 | 6 | 2 | 12 |
| Southern Carpathians = SC CP2 | 46 | 0 | 25 | 3 | 2 | 24 |
| SC-CP3 | 32 | 0 | 36 | 7 | 6 | 19 |
| SC-Total | 35 | 0 | 29 | 7 | 4 | 25 |
| Total Carpathians CP2 | 44 | 1 | 41 | 3 | 1 | 10 |
| Total-CP3 | 33 | 3 | 39 | 8 | 3 | 14 |
| Total-Total | 38 | 1 | 38 | 5 | 2 | 16 |

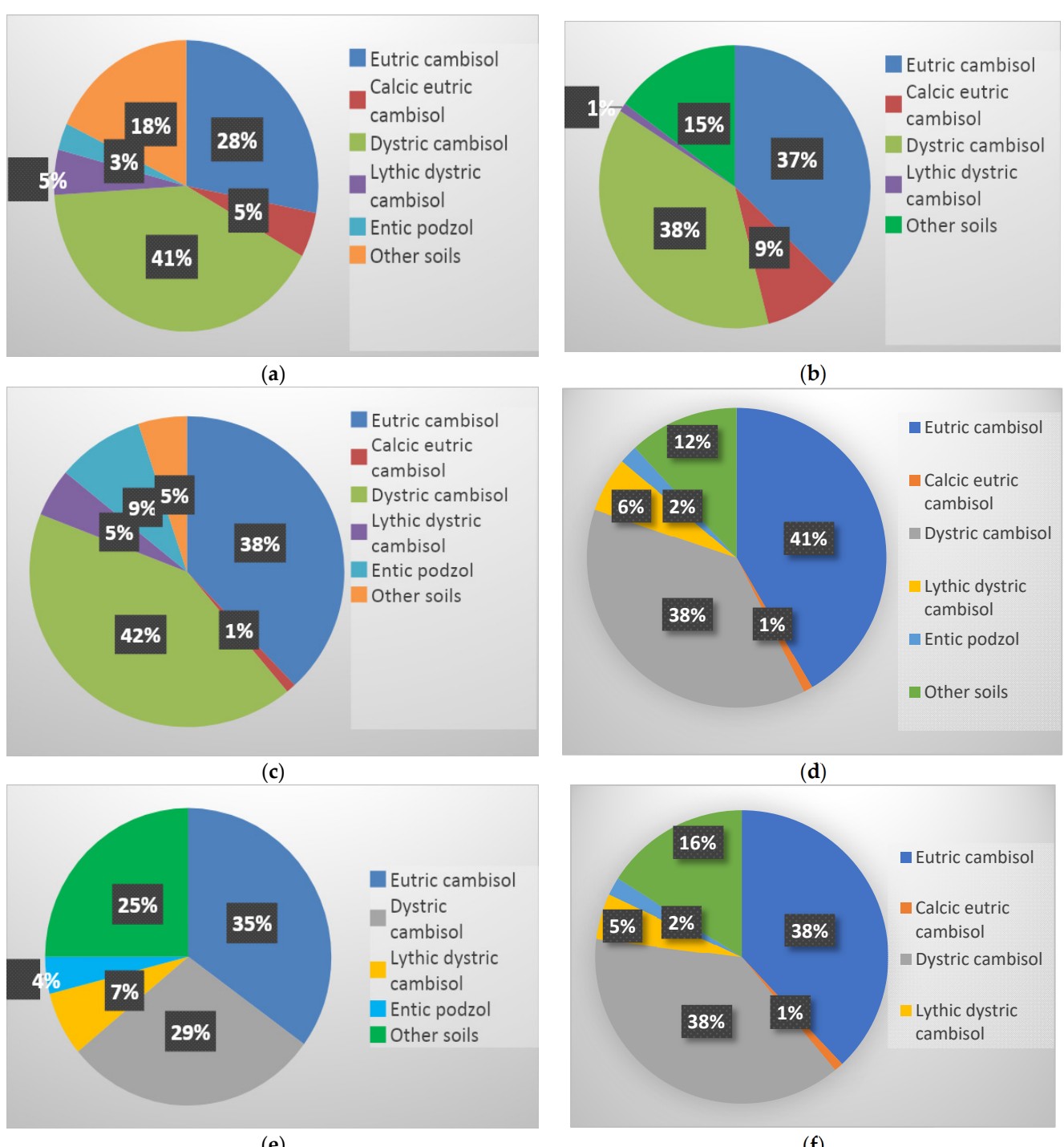

**Figure 5.** (**a**) Soil AM. (**b**) Soil BM. (**c**) Soil EC. (**d**) Soil CC. (**e**) Soil SC. (**f**) Soil Total.

### 3.5. The Participation Percentage of Silver Firs in the Stand Composition

      The participation percentage of the silver tree in the stand's composition is very similar for all five mountain chains (Table 5). The largest percentages are of 10% and 20%, a fact that confirms the predilection of silver fir for mixed stands. Pure silver fir stands (100% composition) are extremely rare.

**Table 5.** Participation of silver fir in the stand compositions from the Carpathians (%).

| Mountain Area | Participation Percentage (%) | | | | | | | | | |
|---|---|---|---|---|---|---|---|---|---|---|
| | **10** | **20** | **30** | **40** | **50** | **60** | **70** | **80** | **90** | **100** |
| Apuseni Mountains = AM CP1 | 44 | 38 | 6 | | 3 | | | | | 9 |
| CP2 | 41 | 27 | 15 | 7 | 4 | 2 | 1 | 1 | 1 | 1 |
| CP3 | 41 | 28 | 15 | 8 | 4 | 2 | 1 | | | 1 |
| CP4 | 47 | 29 | 11 | 6 | 3 | 3 | 1 | | | |
| CP5 | 58 | 32 | 3 | 7 | | | | | | |
| Total | 42 | 28 | 14 | 8 | 4 | 2 | 1 | | | 1 |
| Banatului Mountains = BM-CP1 | 37 | 24 | 13 | 4 | 3 | 3 | 4 | 4 | 6 | 2 |
| CP2 | 40 | 26 | 12 | 7 | 5 | 3 | 3 | 2 | 1 | 1 |
| CP3 | 42 | 26 | 12 | 7 | 5 | 3 | 2 | 1 | 1 | 1 |
| CP4 | 48 | 30 | 8 | 4 | 3 | 3 | 2 | 1 | | 1 |
| CP5 | 27 | 46 | 9 | | | 9 | | | 9 | |
| Total | 41 | 26 | 12 | 7 | 4 | 3 | 3 | 2 | 1 | 1 |
| Eastern Carpathians = EC | 38 | 30 | 17 | 7 | 3 | 2 | 2 | 1 | | |
| CP2 | 45 | 31 | 13 | 6 | 3 | 1 | 1 | | | |
| CP3 | 47 | 31 | 12 | 5 | 2 | 2 | 1 | | | |
| CP4 | 48 | 30 | 13 | 5 | 3 | 1 | | | | |
| CP5 | 44 | 35 | 12 | 5 | 2 | | | | | 2 |
| Total | 45 | 31 | 13 | 6 | 2 | 2 | 1 | | | |
| Curvature Carpathians = CC-CP1 | 21 | 25 | 21 | 12 | 9 | 6 | 3 | | 1 | 1 |
| CP2 | 34 | 31 | 15 | 8 | 5 | 3 | 2 | 1 | | 1 |
| CP3 | 40 | 32 | 15 | 6 | 3 | 2 | 1 | 1 | | |
| CP4 | 44 | 32 | 14 | 6 | 2 | 1 | 1 | | | |
| CP5 | 50 | 34 | 6 | 6 | 4 | | | | | |
| Total | 38 | 31 | 15 | 7 | 4 | 3 | 1 | 1 | | |
| Southern Carpathians = SC-CP1 | 25 | 23 | 17 | 10 | 9 | 8 | 4 | 1 | 2 | 1 |
| CP2 | 35 | 29 | 15 | 8 | 6 | 3 | 2 | 1 | 1 | |
| CP3 | 43 | 34 | 14 | 5 | 2 | 1 | 1 | | | |
| CP4 | 42 | 35 | 14 | 5 | 2 | 2 | | | | |
| CP5 | 37 | 37 | 14 | 5 | 4 | 2 | 1 | | | |
| Total | 40 | 32 | 14 | 6 | 3 | 2 | 1 | 1 | 1 | |
| Total Carpathians-CP1 | 35 | 29 | 17 | 7 | 4 | 3 | 2 | 1 | 1 | 1 |
| CP2 | 41 | 30 | 14 | 7 | 4 | 2 | 1 | 1 | | |
| CP3 | 44 | 31 | 13 | 6 | 2 | 2 | 1 | 1 | | |
| CP4 | 45 | 32 | 13 | 5 | 2 | 1 | 1 | 1 | | |
| CP5 | 42 | 36 | 11 | 5 | 3 | 2 | 1 | | | |
| Total | 42 | 31 | 14 | 6 | 3 | 2 | 1 | 1 | | |

*3.6. Distance from the Road*

The obtained data regarding the distance of silver firs from the road are not relevant: superior productivity stands are closer to existent roads in BM, while the situation is opposite in CC and SC where stands are far-off from roads (Table 6).

**Table 6.** Distance from the road (km) of silver firs located in the Carpathians.

| Mountain Area | Distance from the Road (km) for Productivity Classes | | | | |
|---|---|---|---|---|---|
| | **1** | **2** | **3** | **4** | **5** |
| Apuseni Mountains = AM | 5.63 | 6.78 | 7.69 | 6.84 | 6.71 |
| Banatului Mountains = BM | 5.92 | 6.28 | 9.43 | 10.72 | 9.10 |
| Eastern Carpathians = EC | 7.50 | 8.13 | 8.94 | 8.73 | 7.51 |
| Curvature Carpathians = CC | 7.14 | 7.12 | 7.87 | 10.86 | 6.94 |
| Southern Carpathians = SC | 5.48 | 5.46 | 6.42 | 8.61 | 8.13 |
| Total Carpathians | 7.19 | 7.34 | 7.36 | 8.69 | 7.28 |

*3.7. Stand Structure*

The majority of silver firs from the Carpathians are relatively uneven-aged (Table 7). There are no significant differences between the five analysed areas and between productivity classes.

**Table 7.** Stand structure of silver firs located in the Carpathians (%).

| Mountain Area | Structure | | | |
|---|---|---|---|---|
| | **1** | **2** | **3** | **4** |
| Apuseni Mountains = AM-CP1 | 6 | 60 | 34 | |
| CP2 | 5 | 57 | 38 | |
| CP3 | 9 | 43 | 48 | |
| CP4 | 2 | 33 | 63 | 2 |
| CP5 | | 45 | 55 | |
| Total | 7 | 45 | 47 | 1 |
| Banatului Mountains = BM-CP1 | 5 | 55 | 37 | 3 |
| CP2 | 8 | 46 | 43 | 3 |
| CP3 | 19 | 56 | 24 | 1 |
| CP4 | 7 | 68 | 24 | 1 |
| CP5 | 9 | 55 | 36 | |
| Total | 12 | 51 | 35 | 2 |
| Eastern Carpathians = EC-CP1 | 2 | 55 | 42 | 1 |
| CP2 | 4 | 40 | 55 | 1 |
| CP3 | 7 | 41 | 51 | 1 |
| CP4 | 6 | 42 | 51 | 1 |
| CP5 | 9 | 44 | 47 | |
| Total | 5 | 42 | 52 | 1 |
| Curvature Carpathians = CC-CP1 | 10 | 27 | 58 | 5 |
| CP2 | 10 | 24 | 61 | 5 |
| CP3 | 15 | 25 | 55 | 5 |
| CP4 | 7 | 27 | 59 | 7 |
| CP5 | 13 | 34 | 34 | 19 |
| Total | 12 | 25 | 58 | 5 |

**Table 7.** *Cont.*

| Mountain Area | Structure | | | |
|---|---|---|---|---|
| | **1** | **2** | **3** | **4** |
| Southern Carpathians = SC-CP1 | 9 | 23 | 64 | 4 |
| CP2 | 8 | 26 | 59 | 7 |
| CP3 | 25 | 17 | 53 | 5 |
| CP4 | 19 | 14 | 58 | 9 |
| CP5 | 29 | 20 | 42 | 9 |
| Total | 19 | 20 | 55 | 6 |
| Total Carpathians-CP1 | 4 | 50 | 44 | 2 |
| CP2 | 6 | 36 | 55 | 3 |
| CP3 | 15 | 31 | 51 | 3 |
| CP4 | 11 | 28 | 55 | 6 |
| CP5 | 20 | 30 | 43 | 7 |
| Total | 10 | 35 | 52 | 3 |

The meaning of numbers from Table 7: 1= even-aged stand; 2= relatively even-aged stand; 3= relatively uneven-aged stand; 4= uneven-aged stand

### *3.8. Crown Density*

In the following table (Table 8), we have shown the crown density of the fir groves in the Carpathians. We can notice that most of the trees from the five mountain ranges of the Romanian Carpathians are trees of almost full consistency (0.7−0.9). The trees characterized by a degraded crown density (0.3) are located mainly in stands from EC and Total Carpathians.

Over 40% of the fir trees in the BM, EC and CC are almost full-grown trees, while the full consistency is found in up to 3% of the fir trees (Figure 6). Regarding Total Carpathians, about 40% of the fir trees have a consistency of 0.8 (almost full), while the full consistency is found in less than 2% of the fir trees.

In all cases, the predominant consistency is of 0.8, followed by 0.7 and 0.9 (Table 8).

### *3.9. The Characteristics of Superior Productivity Silver Fir Stands*

By grouping productivity classes in three groups (superior, average and inferior), we obtain similar results with those from points 3.1–3.8. As such, it can be observed that the Carpathian silver fir has superior productivity classes at lower altitudes, on shadowed field aspects (North, North-East and North-West), and on fields with lower slopes. Significant altitude and field aspect differences are found at AM, EC and SC.

These are the most representative mountain chains as surface and geographic position.

The other environment (soil type), stand (composition, structure, consistency) or location characteristics (distance from the road) do not differ significantly based on productivity class.

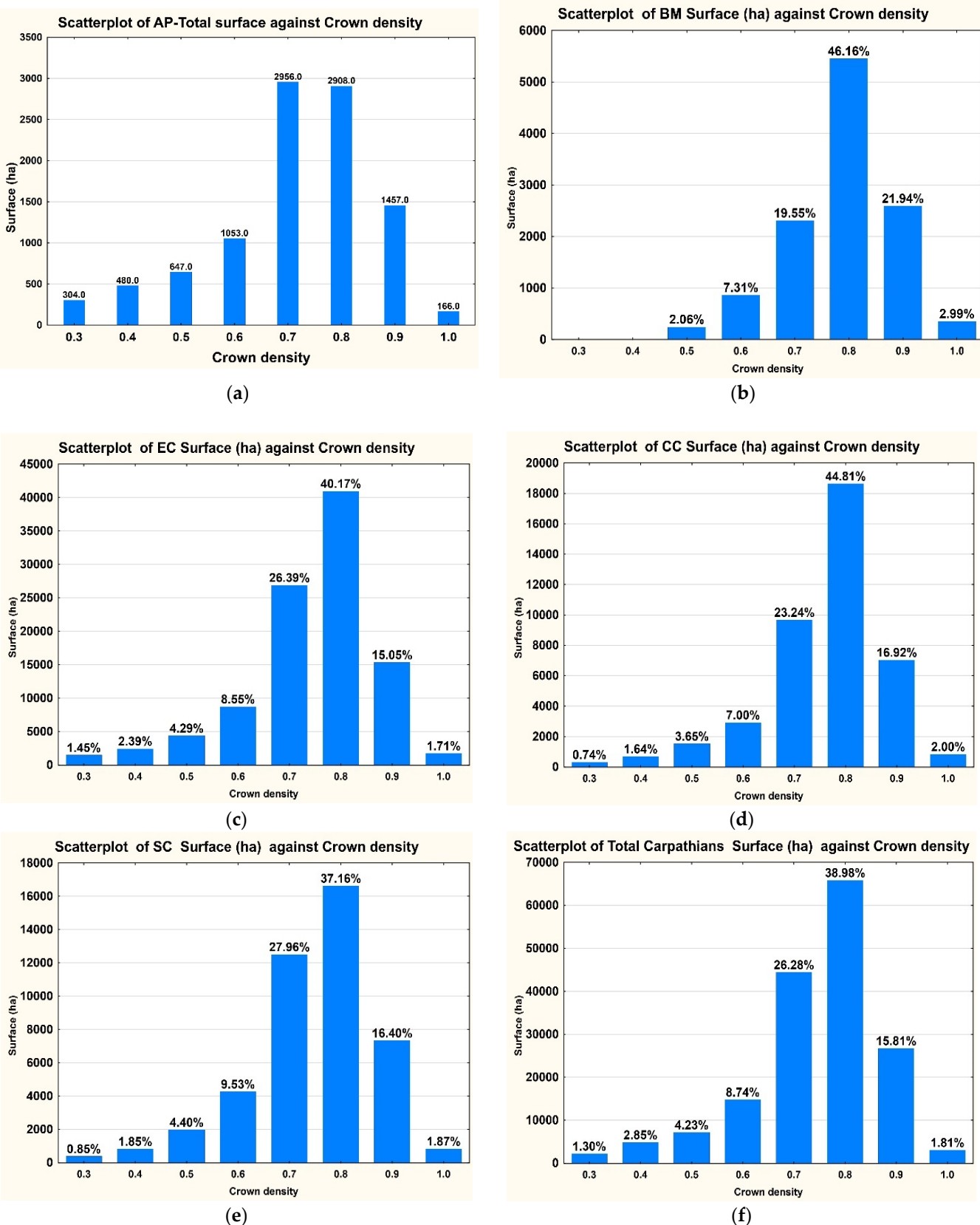

**Figure 6.** (**a**) AM surface according to crown density. (**b**) BM surface according to crown density. (**c**) EC surface according to crown density. (**d**) CC surface according to crown density. (**e**) Crown density SC. (**f**) Total Carpathians surface according to crown density.

**Table 8.** The crown density of silver firs located in the Carpathians (ha).

| Mountain Area | Crown Density | | | | | | | |
|---|---|---|---|---|---|---|---|---|
| | 0.3 | 0.4 | 0.5 | 0.6 | 0.7 | 0.8 | 0.9 | 1.0 |
| Apuseni Mountains = AM-CP2 | | 85 | 179 | 187 | 459 | 679 | 294 | |
| CP3 | 226 | 362 | 374 | 717 | 2283 | 2121 | 1117 | 98 |
| CP4 | | | 81 | 132 | 194 | 86 | | |
| Total | 304 | 480 | 647 | 1053 | 2956 | 2908 | 1457 | 166 |
| Banatului Mountains = BM-Cp1 | | | | | 188 | 427 | 227 | |
| CP2 | | | 172 | 576 | 1301 | 3299 | 1135 | 93 |
| CP3 | | | 50 | 218 | 723 | 1610 | 1139 | 223 |
| CP4 | | | | | 91 | 106 | 90 | |
| Total | | | 243 | 863 | 2309 | 5453 | 2591 | 353 |
| Eastern Carpathians = EC-CP1 | 177 | 240 | 575 | 941 | 4113 | 6393 | 2148 | 291 |
| CP2 | 879 | 1464 | 2412 | 5106 | 15,590 | 25,267 | 8601 | 1091 |
| CP3 | 403 | 663 | 1250 | 2445 | 6672 | 8983 | 4540 | 354 |
| CP4 | | | 124 | 211 | 503 | 328 | | |
| Total | 1480 | 2437 | 4377 | 8725 | 26,913 | 40,974 | 15,353 | 1742 |
| Curvature Carpathians = CC1 | | | | 147 | 415 | 652 | 185 | |
| CP2 | 190 | 263 | 780 | 1610 | 4421 | 8958 | 2530 | 486 |
| CP3 | | 381 | 636 | 1057 | 4345 | 8474 | 4189 | 296 |
| CP4 | | | | 94 | 478 | 531 | 133 | |
| Total | 306 | 684 | 1518 | 2914 | 9669 | 18,640 | 7038 | 830 |
| Southern Carpathians = SC1 | | | 167 | 161 | 523 | 555 | 86 | |
| CP2 | 187 | 268 | 658 | 1591 | 4282 | 6038 | 1912 | 319 |
| CP3 | 163 | 497 | 821 | 1914 | 6072 | 9121 | 5088 | 493 |
| CP4 | | | 270 | 488 | 1399 | | | |
| Total | 379 | 827 | 1968 | 4263 | 12,511 | 16,630 | 7339 | 837 |
| Total Carpathians-C1 | 195 | 277 | 750 | 1156 | 4836 | 7396 | 2491 | 343 |
| CP2 | 1143 | 1884 | 3421 | 7460 | 21,632 | 35,283 | 11,943 | 1553 |
| CP3 | 800 | 2533 | 2495 | 5293 | 15,750 | 21,835 | 11,884 | 1167 |
| CP4 | 53 | 115 | 486 | 853 | 2187 | 1334 | 399 | |
| Total | 2191 | 4809 | 7152 | 14,762 | 44,405 | 65,848 | 26,717 | 3063 |

## 4. Discussion

Due to the tremendous importance of silver fir in providing the manifolds economic, protective and ecologic functions this study focused on assessing the influence of environmental conditions and stand characteristics of silver fir stands from five mountain ranges of the Romanian Carpathians.

### 4.1. Altitude

The present study has shown that silver fir of superior productivity class is located at low altitudes (870−1040 m) in most Carpathian chains from Romania. This fact is justified by the fact that stands located at higher altitudes undergo less intense silvicultural interventions so that their productivity is reduced [21]. Furthermore, these stands fulfil different protective functions [33,45]. As a result, stands located at higher altitudes fulfil

especially the protection function and are mainly covered by special conservation works that are not aiming the increase of forest productivity. This type of management is similar to the forest's natural structure, whose evolution is controlled by attitudes as key factor of the local climate. Although in the case of silver fir, higher altitudes limit the development of this species because of its shadow requirements, this practice ensures a better resistance towards the disturbing factors [46]. On the other hand, silver fir is present especially in common beech mixtures at altitudes between 700 m and 1200 m [34]. A predominance of silver fir mixture stands at altitudes of up to 1100 m is also reported by Klopčič [21]. Mixtures between evergreen and deciduous species favour a more vigorous silver fir growth due to leaf phenology [47].

Silver fir crowns entrap light all along the year, fact that leads to superior productivity classes as was noticed also in our research [48,49]. The influence of some environmental conditions (especially climatic) over the productivity of silver fir stands is also stated by other authors who emphasize that altitude is strictly correlated with climate conditions [16,35–37]. Hilmers [16] has shown that the productivity of silver fir stands increases as temperature grows at high altitudes while Toromani [50] mentions that the extension of the vegetation season favours the increase of silver fir productivity. These projections are confirmed both for the territory of our country and for the Carpathian area, which may confirm the higher productivity of silver fir in some Carpathian chains such as Eastern and Southern parts [51,52]. Moreover, the silver fir productivity increment in stands located at altitudes of up to 1100 m as we observed in our research, is correlated with the temperatures from the cold season (January-March) as other authors have already shown [40]. This situation can be explained by the temperature increase during winter that reduces the risk of frost so that stand productivity is not affected [50]. On the other hand, the soil water deficit is a limiting factor that can increase the silver fir's vulnerability to drought and a decrease of its productivity if it appears in the warm season [48]. However, other authors have shown that, although silver fir stands are located at lower altitudes, those can record inferior productivity classes [48]. This is the case of stands from BM, that although are located in Western Romania, area which is not exposed to could waves, but in which reduced precipitation influence stands productivity due to the reduced water deficit [53].

### 4.2. Field Aspect

Our results regarding the preference of superior productivity silver firs for shadowed field aspects are in accordance with the shadow character of this species, mentioned by many authors [33,53–56]. The growing stock of fir in the northern field aspect exceeds that in the southern field aspect [57–59] because the fir is supplied with a more suitable light and water regime on northern field aspect [60,61]. Other authors have shown that, compared with the sunny areas, the shadowed ones favour an increased tree diversity as this is characterized by optimum temperature and humidity conditions [55]. More than that, silver fir is a species resistant to low temperatures, frosts and high precipitation quantities [11,28,61,62]. On the other hand, water availability is a limiting factor in the growth and development of silver fir. South field aspects are characterized by a higher probability for the appearance of water deficit due to a more accentuated solar radiation [46,55,63]. However, when high temperatures do not overlay with a precipitation deficit, the silver fir growth is favoured, thus leading to higher classes of productivity [50].

### 4.3. Field Slope

Our observation that low productivity silver fir is located, in all cases, on fields with a very high slope is correlated with the fact that those are situated at the highest altitudes. This corresponds in the mountain area with fields near mountain peaks that also have a high slope. As other studies highlighted these fields have a low accessibility degree so that the intervention with silvicultural treatment which sustain silver fir growth is limited [64]. Similar results were also reported by other authors who observed a predisposition of this species for mid to upper and steep gradients, especially when silver fir appears in common

beech and Norway spruce mixtures [65]. This situation can be caused by the fact that fields located on steeper slopes are mainly fields with superficial soils characterized by a low soil water content that are not favourable for silver firs, as it was mentioned in other studies [41].

### 4.4. The Soil

Silver fir prefers acid, deep soils, with a high humidity content [66]. Eutric cambisol and dystric cambisol are the characteristic soils for silver fir. Besides a sufficient depth and humidity, these soils are rich in nutritive substances and are characterized by an intense biologic activity [67–69]. As a consequence, our results show the preponderance of silver fir stands on eutric cambisol and dystric cambisol, a fact that is in line with this specie's ecologic requirements. Other authors have also observed vigorous silver fir growths on acid soils [66]. Tree growth is influenced by soil depth, and thickness of genetic soil horizons [27]. Soils rich in calcium are more favourable for silver fir [15,70,71]. This is the case of calcic eutric cambisol, which has a higher presence in one of the causes of spreading valuable stands in this region [72]. On the other hand, lithic soils are not favourable for silver fir due to its taproot system [34,73,74]. In this contest, silver fir stands from the Southern Carpathians have lower productivities.

### 4.5. Participation Percentage of Silver Fir in the Stand's Composition

The participation percentage of silver fir in the stand's composition depends not only on the local specific (region, soil, humidity conditions) but also on the applied silvicultural system applied [66]. Our study revealed that the largest proportions of silver fir participation in the stand composition are 10% and 20% respectively. This fact confirms the silver fir's predilection for mixed stands, while pure and even aged silver fir stands (composition: 100%) are extremely rare [66]. Unlike pure stands, mixed stands offer a better resilience against harmful factors (biotic and abiotic) and, implicitly, a better phytosanitary stand state [47,75]. In addition, a silver fir participation percentage of up to 20% in the stand's composition contributes significantly to ensuring the sustainability of forest ecosystems [76].

Similar data with our research were also obtained in other studies: the participation of silver fir does not exceed 0.1–0.4 of stand composition in fir-spruce and lime-fir-spruce forests from Vyatka-Kama biome, Russia [77]; the participation of silver fir in mature stand composition was about 20–30% at altitudes between 900 and 1200 m in Eastern Carpathians. Over this altitude the presence of silver fir in stand composition decreases sharply [17]. Furthermore, a high percentage of silver fir in the stand's composition will lead to the decline of this species especially in mountain areas where the effects of climate changes are more evident and where they diminish the vitality of this species [46,75].

These participation percentages can also be connected with the decline of silver fir from the last decades. Numerous similar situations were identified only in the Carpathian area in Western Carpathians from Slovakia [78,79], in the northern Carpathians (Czech Republic, Slovakia), in Slovenia and Croatia [66], and in the Ukrainian Carpathians [80].

The distance from the road is normal to not be a decisive factor in distributing stands on productivity classes. Other authors [81,82] have also observed that the spatial extent of road effects on plant communities from forests remains unclear. However, closeness to a network of existent roads facilitates the appliance of silvicultural treatments that lead stands towards superior productivity classes, as it was obtained in BM. On the other hand, distance from the road and, implicitly, limiting the intervention of silvicultural or exploitation works leads to a decrease of stand productivity [45]. However, we have obtained an inferior productivity in CC for a minimum distance from the road. We consider that this situation is caused by superior altitudes that limits the appliance of silvicultural interventions and favours the protection function against the productivity one.

Stand structure plays a fundamental role in improving tree resilience, especially in the context of climate changes [83]. Furthermore, it is important to know stand structure for understanding forest management [34,84]. In the case of silver fir stands, their characteristic

structure is relatively even-aged and relatively uneven-aged [34,85]. These management practices are economically efficient as they ensure a vigorous tree growth and, consequently superior productivity classes [85]. Many studies have shown that silver fir stands managed in this way are characterized by a better resistance and stability against harmful factors, including $SO_2$ emissions that, in high percentages, increase silver fir mortality rates [83,86]. Nevertheless, some authors mention that a relatively uneven-aged structure favours silver fir better [21,66].

*4.6. Stand Crown Density*

The study performed for resinous stands located in the Southern Carpathians, Romania, has shown that structure, consistency, relief, slope and flora type are the most important factors in defining stand structure [26]. Our results show that in the case of the five studied mountain chains, the majority of silver fir stands (approximately 45%) are stands with an almost full consistency. This observation is in accordance with this specie's temperament, the silver fir being a shadow species [32,66]. Alternatively, full consistency stands occupy very small percentages (2–3%). This type of consistency is not recommended in the case of silver fir stands because even though the species has a shadow temperament, a full consistency would endanger the regeneration's success [53,59,83]. Being mainly found in mixtures, silver fir records higher growth rates when compared with other species. This fact can be attributed to a larger amplitude of silver fir crowns that can occupy more than half of the tree's height [21,66]. Crown length and consistency has a decisive role in the growth of tree vitality, as it is also mentioned by other authors [87]. In addition, stands with an almost full consistency have a better resistance against harmful factors such as mistletoe infection [17]. As a consequence, we can consider that stands with an almost full consistency are favourable to the silver fir's sustained growth which, in these conditions realizes superior productivity classes.

The best conditions where silver fir realizes superior productivity classes are:

➢ lower altitudes up to approximately $1100-1200$ m [26,88];
➢ shadowed field aspects, characterized through better temperature and humidity conditions [46,59];
➢ fields with a slope up to $30°$, which is favourable to a sustained development that favours the superior productivity classes [41,65];
➢ a silver fir's tree participation percentage up to 20% in mixtures with other species [76,77];
➢ stands characterized by an uneven-aged structure [46] and incomplete canopy closure (0.7–0.9) [17].

## 5. Conclusions

There is already evidence that silver fir has been declining at European level due to air pollution and the effects of excessive climates. In this context, changes in the forest management are necessary to improve forest resilience under climate change, particularly considering the silver fir's potential to ensure multiple productive and protective functions.

Our analysis revealed that a higher silver fir productivity class is found on sites located at altitudes of up to 1200 m (AM, EC, SC). Regarding the field aspect, we found that the most suitable sites have NW field aspects (AM, EC and SC) because these ensure the most favourable conditions of light and water compared to the southern field aspects. These findings are in line with the specie's shade temperament. Silver fir growth performances are recorded on mid- to upper and steep slopes, which can be explained through the depth and water content of the soil layer. Another important characteristic that controls silver fir productivity is soil, with eutric cambisol and dystric cambisol being the most favourable for silver fir growth performance in all considered sites. Our study suggests that silver fir participation in the stand composition should be between $10-20\%$ in order to ensure an increased silver fir stand resilience on various disturbing factors. Other fundamental characteristics of silver fir are stand structure and consistency. Hence, our findings revealed that the application of relatively even-aged structure and almost full crown density ensures

not only a vigorous growth of silver fir and implicitly higher productivity classes but also optimum requirements for species regeneration. However, some characteristics, like positioning (distance from the road) of the silver fir trees did not show important differences in the productivity class.

Overall, we can conclude that the fist hypothesis established at the beginning of this study is confirmed, there is a strong connection and a certain dependence between the production classes from the forest areas of different mountain massifs. Regarding the second hypothesis we can assert that altitude, filed aspect, slope, the percentage of tree participation and stand structure are influencing the silver fir productivity class.

The findings of the present study are consistent with numerous previous studies that concluded that a higher growth performance of silver fir is recorded in certain relief-stand conditions. Under these conditions, mainly mixed silver fir stands would ensure the provision of ecosystem services. Despite its increased adaptation potential to climate change, it is fundamental that forest managers pay more attention to these driving factors when designing future strategies that aim towards forest resilience and sustainability.

**Author Contributions:** Conceptualization, L.D. and M.M.; methodology, L.D. and V.R.; software, G.M.; validation, R.D. and R.C.; formal analysis, G.M. and V.R.; investigation, R.D. and L.G.; resources, R.C.; data curation, L.D.; writing—original draft preparation, L.D., M.M. and V.T.-G.; writing—review and editing, L.D., M.M., V.R. and V.T.-G.; visualization, V.T.-G.; supervision, L.D.; project administration, L.D.; funding acquisition, L.D. All authors have read and agreed to the published version of the manuscript.

**Funding:** This paper was carried out through the project "Increasing the institutional capacity and performance of INCDS "Marin Drăcea" in the activity of RDI—CresPerfInst" (Contract no. 34PFE./30.12.2021) financed by the Ministry of Research, Innovation and Digitization through Program 1—Development of the national research—development system, Subprogramme 1.2—Institutional performance—Projects to finance excellence in RDI.

**Institutional Review Board Statement:** Not applicable.

**Informed Consent Statement:** Not applicable.

**Data Availability Statement:** The data presented in this study are available on request from the corresponding author. The data are not publicly available due to the project through which the research was funded is not yet completed.

**Acknowledgments:** The work of Gabriel Murariu was supported by the "An Integrated System for the Complex Environmental Research and Monitoring in the Danube River Area", REXDAN, SMIS code 127065, co-financed by the European Regional Development Fund through the Competitiveness Operational Programme 2014-2020; contract no. 309/10.07.2020.

**Conflicts of Interest:** The authors declare no conflict of interest.

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
