# Peer review of "Which Are the Best Site and Stand Conditions for Silver Fir (Abies alba Mill.) Located in the Carpathian Mountains?"

_diversity, doi:10.3390/d14070547_

Round 1
Reviewer 1 Report
Diversity 2022, 14, …, 2nd review
Article: Which are the best sites and stand conditions for European silver fir (Abies alba Mill.) …
Dina L., Marin M., … Timic-Gansac V.
Comments for the authors
General remarks
Authors followed the suggestions of the reviewers mostly.
Detailed comments
Introduction
page 1, line 44 etc.:
This reviewer is still convinced that it would be better not to emphasize the entire Europe. Suggestion: ‘… mountainous areas of Europe…’ or ‘…mountainous up to sub-alpine landscapes of Europe …’ There are a lot of other tree species outside from mountains in Europe which are also valuable covering huge areas.
Material and Methods
page 3, Figure 1: Letters under ‘Legend’ are still too small and should urgently be enlarged.
Results
page 7, lines 177-178: This reviewer does still not understand what means ‘… due to compensation on areas … ‘. Can this be explained?
page 15, Figure 6: It was forgotten to enlarge the heading of (a) and the texts along all axes everywhere (x- and y-axes).
page 17: lines 305-306: Please delete finally ‘…the occurrence of negative phenomena…’ Is it not self-evident that avalanches are negative phenomena? Should that be mentioned explicitly?
line 321: Hilmers et al. …Is it number 16 in the reference list?
Discussion and Conclusions
page 19 and almost all lines: This reviewer is still be convinced that the list at the end of the sub-chapter ‘Stand crown density’ would better fit to ‘Conclusions’ at least. Both chapters could be combined in one.
References
Only samples were again checked at random.
page 22, line 562: ‘…Mill. …’
Final remark
It is recommended to publish this manuscript after a few additional improvements.
Author Response
We send you in draft the changes made. Thank you very much!

Reviewer 2 Report
Pg.1, Ln 1, Title – I consider the word "European" to be unnecessary, because the name of the tree species is without European, it is not a specific species. In addition, it is clear from the article that the object of the study is a location in Europe.
Pg.1, Ln 17, Abstract and Pg.2, Ln 49, Introduction – correct (Picea abies (L.) Carst) to ……….Karst.
Pg.1, Ln 22 Abstract – which 10 years? Please specified (from ……..to…….).
Pg.1 and 2, Introduction – there is a lack of works on fir revitalisation in some parts of Europe in recent decades. There are also missing papers on the possible influence of different management of fir stands on their diameter increase/decrease in the past and/or given the changed conditions due to the current climate change.
Pg.2, Ln 98-100 – this sentence belongs rather to the conclusions. It needs to be moved there.
Pg.3, Ln 111, Materials and Methods – corrections [92] in bracket instead of [922]. The whole chapter needs to be rewritten and/or adjust. I recommend dividing the chapter into 3 subchapters (Object, Data collection, Statistical analysis).
Pg.3, Ln 115 – 117; this data are incorrect in this form - a comma should be used instead of dots.
Pg.4, Ln 132 and 133; According to which classification was the stand structure and consistency evaluated? Why was the fir proportion in the stands not taken over from the Forest Management Plans, but was evaluated visually? The data in Table 5 indicate that the proportion of fir proportion in the stands was calculated, in other words, how can the proportion be determined visually with the accuracy of 1%?
Pg.4, Ln 137 ; What do the classes of productivity 3 + 4 + 5 mean? It needs to be explained, also in the methodology.
Pg.4, Ln 147-150; working hypotheses move to the end of the Introduction chapter for work objectives.
Pg.4 to 16 , Results, all Table and Figures lack an explanation, what are productivity classes? Please, define it also in the methodology.
Pg.13, Table 7 - missing classification what does structure 1 to 4 mean? Please, define it also in the methodology.
Pg.20, Conclusions – I recommend adding whether or not the working hypotheses defined on page. 4 were confirmed.
Author Response
We send you in draft the suggested changes.
Thank you very much !

This manuscript is a resubmission of an earlier submission. The following is a list of the peer review reports and author responses from that submission.
Round 1
Reviewer 1 Report
Diversity 2022, 14,...
Article: Which are the best site and stand conditions for European silver fir (Abies alba Mill.) ...
Dina L., Marin M., ....Timic-Gansac V.
Comments for the authors
General remarks
This abundant and comprehensive study is especially valuable because it reflects a wide spectrum of knowledge and experiences about the performance and environmental demands of Abies alba being one of the dominant tree species in one of the most important mountainous areas in Europe, the Carpathians.
Based on on large data sets it contributes new aspects which are useful for plant ecology and forestry.
Nevertheless, this manuscript needs some improvements before publication which on the one hand concerns the scientific content and on the other hand some more or less formal problems (Compare: `Detailed comments'!).
Detailed comments
Abstract
page 1
lines 13-14: These two sentences should be omitted, because they contain common knowledge which must not permanently be repeated. Recommendation: Start with `...Silver fir (Abies alba Mill.) ...'
line 15: '... across European mountains ...' It does not occur in flat areas of low land.
line 24: This reviewer thinks that it would improve the 'Abstract' if authors could list all 8 factors here already because abtsracts are often read separately.
line 27: What means '... relatively even-aged stands ...'? It sounds as if theses stands were planted.
Introduction
page 1
line 42 and line 45: It is recommended to talk of mountainous areas of Europe, because compared to for instance Pinus silvestris it is a rather rare species if one considers the whole Europe, and it became progessively rare during the last decades (in mountainous up to sub-alpine landscapes).
page 2
line 77: '... Viscum album ssp. abietis ...' (lowercase letter !)
line 86: ?? '... On the other stands ...' Should it not be '...On the other hand ...'?
Material and Methods
page 3: Figure 1; The letters in the legend are too tiny. Could you enlarge them?
Results
page 4-5
Figure 2: Again the letters are too small and might not be visible after printing.
This comment has to be repeated for the figures 3 and 4-6. It is suggested to overcome this problem by shrinking the size of all tables in favor of the figures.
page 6
Figures 3 (a) and (b) ; Lines and letters are too tender.
page 7
line 172: '... Silver ...' (a rich land!), probably Siver fir!
line 177: This reviewer does not understand the meaning of '..., due to compensation on areas...'
page 7-8
Figure 4: Compare the comment above!
page 9
Figure 5: Unfortunately, the nice graph shows some overlapping numbers in (d) and (f) Please improve that!
page 10
line 208: '... 3rd ...', r
line 211: '... and 3rd...', r
page 12
line 219: '... siver tree...' ? instead of siver fir ??
page 13
lines 234-235: Can those firs be at the same time be 'relatively even-aged' and relatively uneven-aged'. There is perhaps a contradiction. Sense?
page 14-15
Figure 6: Letters and numbers are too small.
page 15
line 258-259: '...p productivity class ...' ? Meaning?
Discussion
page 16
line 270: ' as well as from other parts of Europe ...'
lines 275-276 '... species ...'
lines 287-288: ' ... a barrier against the occurence of different negative phenomena such as landslides and avalanches.' Delete 'the occurence of ... '
line 308: ' ... increment ...'
page 17
line 318: ' ... in agreement ...' or ' ... in accordance ...'
line 329: ' ... to the of (?) superior productivity ...' Meaning?
line 336: ' ...mid -... Minus not necessary.
line 339: ' ... implicitly for the apparition of superior productivity classes.' Sense?
line 354: ' ... pivotant ... ' ? Probably French not English.
page 18
lines 376-377: Two verbs in one sentence? ' ...are ...'? Is 'area' meant?
Lines 395-396: CO2 cannot increase mortality but on the contrary is needed for photosynthesis. Perhaps it should be '... SO2 .... '?
page 18-19
lines 417-428: 'What are the best conditions ... etc. ' This completely belongs typically and exclusively to 'Conclusions'.
line 423: Authors contradict themselves. In line 351 they say that silver fir prefers soils rich in calcium (This is also the opinion of this reviewer.) And now they say that silver fir prefers acid soils. There is some contradiction! (pH was not measured in this study.) Ca rich soils are not acid.
Conclusions
Please connect the last part of 'Discussion' with 'Conclusions'
Refeferences
Samples were taken at random only. Please check everyting!
page 20
line 513; ' ... abies ...'
page 21
lines 538-539: Why suddenly capitals (upper-case charakters)? Please try to stay uniform!
line 578: '... Abies alba Mill. ...'
page 22
line 605: Why '... Forest ...' with capital?
lines 614-615: Why capitals?
line 631: '... alba Mill. ...'
page 23
line 642: '... Anethum graveolens ... In italics please!
line 658: CO2 subscript 2!
lines 661-662: Why capitals? Try to remain uniform!
Line 665: ' ... in a B ... ? Meaning?
Line 668: ' ... (Abies alba Mill.)
page 24
line 101: This reviewer does not understand what this listing of names means.
Final remark
This manuscript needs to be improved in some parts. After that the revised version can be published.
Reviewer 2 Report
Several typographic errors have been observed. An orthographic revision (spelling checker) is recommended, after a major review of the work, collected in the attached file.

Reviewer 3 Report
By the authors, the objectives of this study is to identify the key characteristics (stational or stand) that lead to the occurrence of superior productivity classes in silver fir from the five Romanian Carpathians mountain chains. In addition, they mentioned in summary that this study investigates the environmental conditions and stand characteristics, based on the data recorded over a period of 10 years. So I was very interested in this paper. Unfortunately, however, I couldn't understand the paper immediately because the terminology used was not appropriate, I think. For example, "participaption percentage" means "species dominance", doesn't it? I think the other terms used in this paper would not be adequate. There is more important issue. It is questionable whether the factors analyzed in this study are essential for assessing the productivity of Silver fir. Even if the soil type, altitude, slope are understandable, the other items are ambiguous in relation to forest productivity. The authors said that the data used in this paper is published in the literature of Forest management plan, but the explanation is insufficient to understand the data contents. What kind of information is contained in the document? It is unclear whether the information is valid as factors to evaluated forest productivity. Regarding the site index, in Line 114-116, this paper states "For uneven-aged stands, this value can be determined by the average height that corresponds to the average diameter of 50 cm." However, the diameter is strongly influenced by the stand density, so that it is not appropriate to evaluate the site index using this value. Also, the proportion of evenaged stands and unevenaged stands in the regions studied may have significant impacts on the average productivity values they evaluated. I couldn't understand what the authors mean in lines 131-133. Section: Distance from the road Sorry, I cannot believe that forest productivity should is affected by ther distance from the road. Section: Stand structure If the "participaption percentage" almost similar to the concept of species composition, the "stand structure" which is denoted as age-related composition of silver-fir by the authors might be represented by "participation percentage", I guess. Namely, other species would gradually invade into uneven-aged silver-fir stands. In other wors, I don't see what the difference between "participation percentage" and "stand structure" is as to stand structure or species composition. Section: Crown density Table 8: I think the numbers of 0.3-1.0 are crown density. Then, what are the numbers in the table (85, 179, 187 and so on)? Figure 6: there is no explanation about "surface". I have no idea about this section. Section "the characteristics of superior productivity silver fir stands" and Discussion. The authors tried to summarize the paper by this section. However, judging from the contents of the results and the way of their explanation, this section is quite unreasonable. Specific Comments I wonder if the terminology the authors used is adequate or not in the context of the paper. For example, participation percentage (dominance?), predilection (niche?), consistency (?? coexistence?), Figure 5: It's better to use the same color for the same thing. Table 4: what are CP1 to CP5? I guess there are no explanation for them.